# The global burden of Chikungunya fever among children: A systematic literature review and meta-analysis

Doris K. Nyamwaya[1,2]*, Samuel M. Thumbi[3,4,5], Philip Bejon[1,2], George M. Warimwe[1,2], Jolynne Mokaya[1,2]

1 KEMRI-Wellcome Trust Research Programme, Kilifi, Kenya, 2 Centre for Tropical Medicine and Global Health, University of Oxford, Oxford, United Kingdom, 3 Paul G Allen School for Global Health, Washington State University, Pullman, Washington, United States of America, 4 Institute of Immunology and Infection Research, University of Edinburgh, Edinburgh, United Kingdom, 5 Center for Epidemiological Modelling and Analysis, Institute of Tropical and Infectious Diseases, University of Nairobi, Nairobi, Kenya

* dnyamwaya@kemri-wellcome.org

**Data Availability Statement:** All data underlying our findings is provided with the submission.

**Funding:** This work was commissioned by the National Institute for Health Research (NIHR)

## Abstract

Chikungunya fever (CHIKF) is an arboviral illness that was first described in Tanzania (1952). In adults, the disease is characterised by debilitating arthralgia and arthritis that can persist for months, with severe illness including neurological complications observed in the elderly. However, the burden, distribution and clinical features of CHIKF in children are poorly described. We conducted a systematic literature review and meta-analysis to determine the epidemiology of CHIKF in children globally by describing its prevalence, geographical distribution, and clinical manifestations. We searched electronic databases for studies describing the epidemiology of CHIKF in children. We included peer-reviewed primary studies that reported laboratory confirmed CHIKF. We extracted information on study details, sampling approach, study participants, CHIKF positivity, clinical presentation and outcomes of CHIKF in children. The quality of included studies was assessed using Joanna Briggs Institute Critical Appraisal tool for case reports and National Institute of Health quality assessment tool for quantitative studies and case series. Random-effects meta-analysis was used to estimate the pooled prevalence of CHIKF among children by geographical location. We summarised clinical manifestations, laboratory findings, administered treatment and disease outcomes associated with CHIKF in children. We identified 2104 studies, of which 142 and 53 articles that met the inclusion criteria were included in the systematic literature review and meta-analysis, respectively. Most of the selected studies were from Asia (54/142 studies) and the fewest from Europe (5/142 studies). Included studies were commonly conducted during an epidemic season (41.5%) than non-epidemic season (5.1%). Thrombocytopenia was common among infected children and CHIKF severity was more prevalent in children <1 year. Children with undifferentiated fever before CHIKF was diagnosed were treated with antibiotics and/or drugs that managed specific symptoms or provided supportive care. CHIKF is a significant under-recognised and underreported health problem among children globally and development of drugs/vaccines should target young children.

Global Health Research programme (16/136/33; funding to GMW) using UK aid from the UK Government. The views expressed in this publication are those of the authors and not necessarily those of the NIHR or the Department of Health and Social Care. Additional support came from an Oak foundation fellowship to GMW, and a Wellcome Trust grant (grant number 203077_Z_16_Z) to PB. The funders had no role in study design, data collection and analysis, decision to publish, or preparation of the manuscript.

**Competing interests:** The authors have declared that no competing interests exist.

## Introduction

Chikungunya fever (CHIKF) is an acute febrile illness caused by the mosquito-borne chikungunya virus (CHIKV) [1]. CHIKV is a positive-sense single-stranded RNA virus of approximately 11 kilobases that is mainly transmitted to humans by *Aedes aegypti* and *Aedes albopictus* mosquitoes [2, 3].

There are three main CHIKV genotypes, the west African (WA) , the east central and southern African (ECSA) and the Asian genotypes [4]. The strains of the WA lineage are maintained in a sylvatic cycle in western Africa countries with small focal spill-over outbreaks of human infections reported [5, 6]. The ECSA strains are enzootic in East, Central and Southern Africa but they have been detected in samples from a broader geographical range [7]. Mutations in the ECSA genotype resulted in the Indian ocean lineage (IOL) [7].

The first CHIKF outbreak was reported in Tanzania (1952) and was followed by several sporadic and major epidemics in tropical and subtropical regions; Africa, Asia, Europe, the Pacific islands and the Americas [8]. Epidemics were mostly experienced in 1960s and 1990s but the 2004 CHIKV epidemic (caused by genotype ECSA), emerged in coastal Kenya and is the largest on record reporting over 1.3 million cases [5, 9]. It spread along the Kenyan coast from Lamu to Mombasa and islands on the Indian Ocean [10, 11], India and subsequently to Southeast Asia and Europe [5, 12, 13]. This rapid expansion into new ecological niches was enhanced by adaptation of CHIKV to new mosquito vectors, its introduction into temperate regions [14, 15] , as well as travel-associated importation of the virus into naïve populations [13] resulting in outbreaks associated with high morbidity and mortality [16].

CHIKV is inoculated by an infected mosquito bite either on human resident dermal cells i.e. fibroblasts and macrophages, or directly unto the blood circulatory system [17, 18]. Initial replication takes place in these skin cells triggering an immune response. It then disseminates to draining lymph nodes for further replication, and spread to other peripheral organs including muscle, peripheral joints, and tendons [19]. In severe cases, the virus invades the brain and liver [19, 20].

CHIKF is characterised by a sudden rise in body temperature and debilitating joint pain that may resolve within weeks or persist for months to years [21]. Other symptoms include; headache, maculo-papular rash, fatigue, myalgia, backache and tachycardia [5, 22]. Severe complications like encephalitis, myocarditis, kidney dysfunction, hepatitis, oculitis, cardiovascular and respiratory disorders have been observed, and are more common among the elderly, infants and immunosuppressed individuals [21, 23]. There are currently no specific antivirals or approved vaccines for CHIKV infection, though efforts towards vaccine development are underway [24–28].

Accurate diagnosis of CHIKF remains a challenge due to non-specific symptoms that are similar to those of other common endemic illnesses such as malaria and dengue [29]. In addition, the general lack of awareness of the disease as well as lack of resources impedes efforts towards active surveillance and inclusion of CHIKF in routine screening of febrile illness. As a result, the burden of CHIKF remains underestimated, with diminished attention on its public health impact [29].

There are few studies that have described the clinical manifestations of paediatric CHIKF, which tend to differ from manifestations described in adults [30–33]. After an incubation period ranging 1 to 12 days, there is a sudden onset of high-grade fever. In adults, this is usually preceded by musculoskeletal manifestations including myalgia, back pain and symmetric arthropathy affecting distal joints that can be sometimes chronic [33]. In contrast, children exhibit a broader range of cutaneous manifestations like pigmentation, bullous rash and blistering [34]. Neurological manifestations including seizures, encephalopathies and

meningoencephalitis are common among children. Haemorrhagic manifestations associated with thrombocytopenia and lymphopenia are commonly reported in paediatric CHIKF [35]. With delayed medical attention, infected neonates have increased risk of developing severe complications including status epilepticus and multiorgan failure due to sepsis and intracerebral haemorrhage [36]. Nevertheless, there remains limited descriptions of the clinical spectrum and epidemiology of paediatric CHIKF, particularly in Africa. We therefore set out to describe the epidemiology of CHIKF among children (aged <18 years) globally by describing the proportion of acutely unwell children who tested positive for CHIKV, geographical distribution, and clinical manifestations through a systematic literature review and meta-analysis.

## Methods

### Search strategy and selection criteria

Our systematic review protocol was developed in line with the Preferred Reporting Items of Systematic Review and Meta-Analyses protocols (PRISMA-P). We searched published peer-reviewed literature in PUBMED, SCOPUS, EMBASE and Web of Science electronic databases for relevant articles without date or language restrictions. We divided search terms into group A; those defining studies in children and group B; terms that define studies focusing on Chikungunya infection as detailed in Table 1. Searches were done for individual groups using the Boolean operator 'OR' and a combined search done for groups A and B using the 'AND' operator as follows: #1 AND #2. PRISMA checklist is provided in (S1 Checklist).

Two researchers (DKN and JM) independently screened titles and abstracts matching the search terms and included studies that reported on laboratory-confirmed CHIKV infection among children (age <18 years), and that presented original data and had undergone peer review. We carried out initial screening of titles and abstracts using the Ryyan software (https://rayyan.ai). Disagreements in the eligibility assessment were resolved through consensus between the two researchers or discussion with a third review author (SMT and/or GMW).

### Data extraction and analysis

Following the selection of eligible studies, DKN and JM independently extracted relevant data using a uniform data extraction tool in Google Sheets (S1 Table). For each study, we extracted the following information: first author and year of publication, study site (country/ geographical location), study setting (hospital vs community), duration of study, whether study was conducted during epidemic or non-epidemic season, study design, sampling method, sample size, diagnostic technique, diagnostic tool sensitivity and specificity, case definition, type of study population, age of study participants, sex of study participants, patient symptoms, clinical/

**Table 1. Details of search strategy used to identify studies on epidemiology of chikungunya infections among children globally, from PubMed, Scopus, Embase and Web of Science databases.**

| Group A: Problem (#1) | Group B: Population (#2) |
|---|---|
| Chikungunya OR "Chikungunya fever*" OR "Chikungunya virus" OR "Chikungunya infection" OR "Chikungunya fever virus" OR "Chikungunya virus infection*" OR "Chikungunya fever* virus infection" OR "CHIKV" | infant* OR infancy OR newborn* OR baby* OR babies OR neonat* OR preterm* OR prematur* OR postmatur* OR child* OR schoolchild* OR "School age*" OR preschool* OR kid OR kids OR toddler* OR adoles* OR teen* OR boy* OR girl* OR minors* OR pubert* OR pubescen* OR prepubescen* OR paediatric* OR paediatric* OR peadiatric* OR "Nursery school*" OR kindergar* OR "Primary school*" OR "Secondary school*" OR "Elementary school*" OR "High school*" OR highschool* |

laboratory findings, treatment, disease outcome, detected viral strain, estimated prevalence/incidence and a statement on the key findings. Non-English language studies were translated into English using *Google Translate* before data extraction.

We used the Joanna Briggs Institute Critical Appraisal tool checklist to assess for quality of case reports (joannabriggs.org) (S2 Table). For assessment of quantitative studies and case series, we used the National Institute of Health quality assessment tool for Observational Cohort and Cross-Sectional Studies (S3 Table), and quality assessment tool for Case Series studies (S4 Table), respectively (https://www.nhlbi.nih.gov/health-topics/study-quality-assessment-tools).

We summarised the clinical manifestations, laboratory diagnostics, treatment administration and disease outcome of CHIKF among children. We estimated the proportion of children with acute CHIKF by pooling all studies with a defined sample size and study population, that screened participants for CHIKF using reverse-transcriptase polymerase chain reaction (RT-PCR) to detect viral RNA in body fluids and/or serologic tests for CHIKV specific immunoglobulin M (IgM). We performed meta-analysis of reported CHIKF among children using the *metaprop* package in R [37]. Overall estimates were calculated using random effects model to take account of between-study heterogeneity [38]. The restricted maximum likelihood estimator [39] was used to calculate the heterogeneity variance $\tau^2$ while the Knapp-Hartung adjustments [40] were used to calculate 95% confidence intervals (CI) around the pooled effect. CHIKF prevalence data were pooled using logit-transformed proportions in a generalised linear mixed-effect model (GLMM). Test for heterogeneity was applied using the Cochran's Q, $I^2$, and H statistics, with an $I^2$ of more than 75% indicating substantial heterogeneity [41]. We performed sensitivity analysis by checking for outlier and influential cases [42] to determine their impact on the validity and robustness of the effect size estimate. We performed meta-regression using multimodel inference to obtain a comprehensive look at which predictors were more or less important for predicting differences in effect sizes. The test statistics and 95% CIs were computed using Knapp-Hartung adjustment. For model evaluation, we applied corrected Akaike's information criterion (AICc). We did not consider any interactions between included predictors (S5 Table). We checked for existence of publication bias by using a funnel plot whose asymmetry was measured by Egger's linear regression test (p<0.05 levels were considered statistically significant for publication bias) [43].

## Results

Our search identified 2104 unique articles. After screening the titles, abstracts and full texts, 142 articles met the inclusion criteria; following exclusion of articles that; did not focus on children, did not describe acute CHIKV infections, were inaccessible, reviews, conference abstracts and general outbreak descriptions (Fig 1). Most studies (32.4%) were hospital-based descriptions of individual cases from an epidemic period (Table 2 and S1 Table). The geographical and temporal distribution of studies included in this review are summarised in Fig 2 and S1 Fig. Most of the studies were from India (n=48) and Brazil (n=17) with only eight countries in Africa represented. There were a number of studies from the Indian Ocean Islands, especially from La Reunion (n=10), a constituent part of France that was severely affected by a CHIKF epidemic in 2005 – 2006 [44]. There were 11 studies from Europe and North America, nine of which were reports on traveller-associated importation of CHIKV [45–53]. Studies from the Carribean islands and Central America,(n=16) were mainly descriptive of epidemic waves after introduction of CHIKV into the presumably immunologically naïve population [34, 54–68]. Case series and case reports represented 76% of the studies from South America.

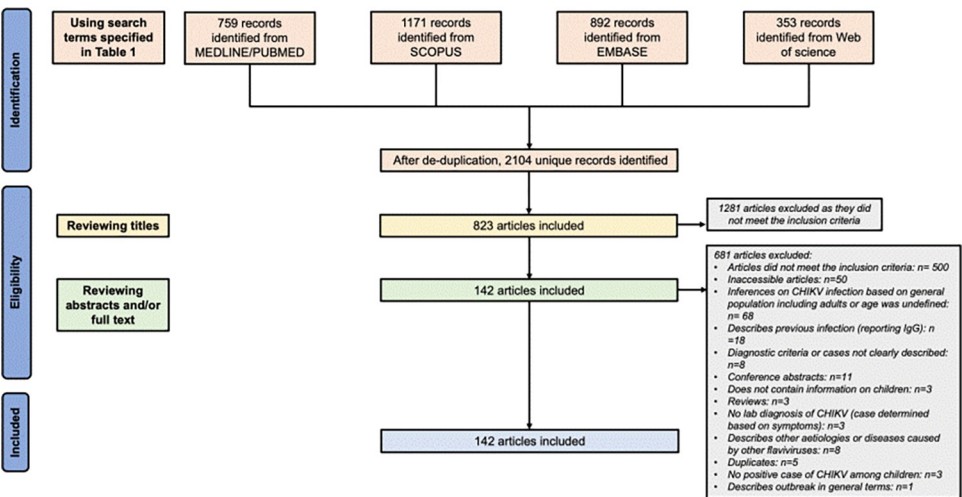

**Fig 1. Preferred Reporting Items for Systematic Reviews and Meta-Analyses (PRISMA) flow diagram illustrating identification and inclusion of studies for a systematic review of Chikungunya disease among children.**

Among the case reports and observational studies and case series, 18 studies [32, 47, 66, 69–83], seven studies [54, 57, 61, 64, 84–86], and one study [87], were of high quality respectively, while the remaining studies had a fair rating (S2–S4 Tables).

To determine the proportion of children with acute CHIKF, we focused on studies (n=53) that had a defined numerator (number of CHIKF cases) and denominator (the total sample size) (S1 Table). We excluded case-reports, case series and observational studies that only reported on children with CHIKF; these studies did not sample from a wider population of individuals with and without the disease. Prevalence rates of CHIKV among children in endemic regions range approximately 5–40% [29, 88–90]. The highest rates were recorded during epidemic seasons (Table 3) with highest positivity observed in the Carribbean islands at 56% (154/275).

Higher estimates of clinical CHIKF are reported among older children [61, 91, 92]. However, the younger children may lack the verbal ability to describe clinical symptoms like myalgia and joint pains. In the absence of effective diagnostics, most cases in this age group remain unrecognized or are misdiagnosed.

Detection of CHIKV RNA by RT-PCR was a frequently used diagnostic method (n=22/53 studies) followed by detection of anti-CHIKV IgM antibodies by ELISA (n=19 studies). Most studies (27/53 studies) were carried out during an epidemic season, and they provided the highest prevalence estimate at 18% (95% CI 10-30, n=27) when compared to those carried out during the non-epidemic seasons, 3.86% (95% CI 2-8, n=6) (Table 4). The percentage of children with acute CHIKF was highest in South America at 22.8% (95%CI 7.5, 51.9) while the percentage in Asia and Africa were comparable at approximately 9% (Table 4). The generated funnel plot was symmetrical implying an absence of evidence of publication bias as suggested by Egger's linear regression test (Intercept= -0.438 [95%CI; -3.33–2.45], t-value= -0.297, p-value= 0.77) (Fig 3).

CHIKF among children can affect various body organs as shown in Fig 4. Fever, rash and arthralgia were the most common symptoms reported in 80, 55 and 40 studies, respectively. The clinical presentation of paediatric CHIKF varies with geographical location and the circulating strain of CHIKV. In La Reunion island where epidemics were predominantly due to the IOL, majority of infected neonates had thrombocytopenia , deranged coagulation, seizures ,

**Table 2. Summary of studies included in a systematic literature review to collate evidence for the global prevalence and clinical manifestation of chikungunya infection among children.**

|  | Number of studies (%) |
|---|---|
| **Study design** | |
| Case reports | 55 (38.7) |
| Case series | 18 (12.7) |
| Cross-sectional studies | 36 (25.4) |
| Longitudinal studies | 29 (20.4) |
| Surveillance | 3 (2.1) |
| Unspecified | 1 (0.7) |
| **Regions represented** | |
| Africa | 24 (16.9) |
| Asia | 54 (38.0) |
| Europe | 5 (3.5) |
| North and Central America | 19 (13.4) |
| South America | 32 (22.5) |
| Oceania | 1 (0.7) |
| Unspecified | 7 (4.9) |
| **Study Setting** | |
| Hospital/health centre | 111 (78.2) |
| Community | 20 (14.1) |
| Unspecified | 11 (7.7) |
| **Season** | |
| Epidemic | 59 (41.5) |
| Non-epidemic | 8 (5.6) |
| Epidemic/Non-epidemic | 1 (0.7) |
| Unspecified | 74 (52.1) |
| **Sex** | |
| Female only | 11 (7.7) |
| Male only | 25 (17.6) |
| Both female and male | 49 (34.5) |
| Unspecified | 57 (40.1) |
| **Case definition** | |
| IgM positive | 36 (25.4) |
| CHIKV RNA positive | 31 (21.8) |
| IgM positive and CHIKV RNA positive | 9 (6.3) |
| IgM positive or CHIKV RNA positive | 18 (12.7) |
| Unspecified | 48 (33.8) |
| **Age categories** | |
| Infants (<1 year) | 47 (33.1) |
| Children (1-12 years) | 14 (9.9) |
| Teenagers/adolescents (13-18 years) | 10 (7.0) |
| Mixed (0-18 years) | 68 (47.9) |
| Unspecified | 3 (2.1) |
| **Viral strain** | |
| Asian | 1 (0.7) |
| West African | 0 (0) |
| ECSA | 11 (7.7) |
| Unspecified | 130 (91.5) |

CHIKV: Chikungunya virus. IgM: Immunoglobulin M. ECSA: East/Central and South Africa

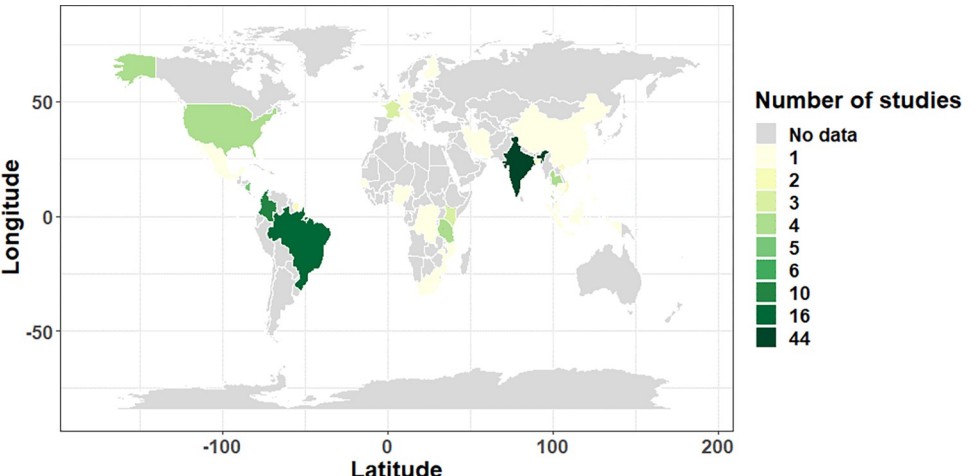

**Fig 2. A global map depicting the geographical distribution of reviewed articles for CHIKV infection in children.** The map was generated using open-source data in the R statistical package "spData" (https://cran.r-project.org/web/packages/spData/index.html). Despite the absence of data from most parts of the globe, most of the studies were from Asia and South America.

hemodynamic disorders and hemorrhagic syndromes [93]. In contrast, circulatory failure was common among CHIKV infected infants in Kerala where the ECSA strain circulated [94]. The Asia genotype introduced to the Carribbean islands and central America induced weaker pro-inflammatory effect with a milder clinical course among infected neonates. It was not associated with thrombocytopaenia and clinical neurological sequelae, however the case fatality rate was comparably high [68, 95].

Of 59 studies that reported on laboratory findings among children with CHIKF, thrombocytopenia was the most commonly observed in 40 studies. Other laboratory findings also reported included: neutrophilia, leukopenia, lymphopenia, hypokalemia, hyponatremia, hypocalcemia, hyperbilirubinemia, anaemia, elevated C-reactive protein, transaminases (aspartate, alanine and oxaloacetic glutamic), elevated cellularity and neutrophil predominance, elevated creatinine kinase, reduced prothrombin concentration (Fig 5). Cerebrospinal fluid (CSF) examination was reported in nine studies [68, 71, 73, 96–100], and showed pleocytosis (n=7 studies), increased protein levels (n=7 studies) and hypoglycorrhachia (n=3 studies). Radiological, CT and MRI scans presented abnormal findings indicating diffuse brain lesions, cerebral edema, soft tissue resorption and haemorrhagic leukoencephalopathy.

Sixty-five of one hundred forty-two studies described treatment therapies for CHIKF. Antibiotic prescription was predominantly given to children presenting with undifferentiated fever

**Table 3. The rates of CHIKV cases reported among children from various continents in different seasons.**

| Continent | Season n/N (%) CHIKV Cases | | | |
|---|---|---|---|---|
| | Both | Epidemic | Non-epidemic | unspecified |
| Africa | 32/383 (8.36) | 516/4729 (10.91) | 124/2011 (6.17) | 43/749 (5.74) |
| Asia | - | 425/1954 (21.75) | 31/1915 (1.62) | 1087/14534 (7.48) |
| C. America | - | 96/3377 (2.84) | - | - |
| Carribean | - | 154/275 (56) | 74/1390 (5.32) | - |
| Indian ocean islands | - | 224/2959 (7.57) | - | - |
| Oceania | - | 13/33 (39.39) | - | - |
| S. America | - | 83/205 (40.49) | - | 194/641 (30.27) |

**Table 4. Results of sub-group analysis for prevalence of CHIKV infection among children based on continent, study setting, diagnostic technique, study design and season.**

| | Subgroup analyses | | |
|---|---|---|---|
| | Number of studies | Proportion in % (95% CI) | I², % |
| **Continent** | | | |
| Africa | 17 | 8.86 [4.42; 16.98] | 97.2 |
| Asia | 20 | 8.78 [3.87; 18.70] | 97.8 |
| North America | 6 | 22.20 [4.22; 64.89] | 99.3 |
| South America | 10 | 22.83 [7.50; 51.88] | 92.5 |
| **Study setting** | | | |
| Hospital/health centre | 37 | 13.02 [7.36; 21.99] | 98.1 |
| Community | 11 | 10.11 [4.38; 21.64] | 98.1 |
| Unspecified | 5 | 7.52 [0.68; 49.25] | 91.3 |
| **Diagnostic technique** | | | |
| Detection of anti-CHIKV IgM | 19 | 11.09 [6.11; 19.31] | 97.3 |
| Detection of CHIKV RNA | 22 | 14.64 [6.64; 29.27] | 97.7 |
| Detection of anti-CHIKV IgM and CHIKV RNA | 12 | 8.56 [2.42; 26.12] | 98.4 |
| **Study design** | | | |
| Longitudinal | 22 | 11.84 [5.14; 24.96] | 98.5 |
| Cross-sectional | 21 | 8.75 [4.65;15.88] | 97.4 |
| Unspecified | 10 | 21.08 [8.46; 43.55] | 95.5 |
| **Season** | | | |
| Epidemic | 27 | 18.46 [10.52; 30.37] | 98.4 |
| Non-epidemic | 6 | 3.86 [1.91; 7.62] | 89.7 |
| Epidemic/Non-epidemic | 1 | 8.36 [5.97; 11.58] | NA |
| Unspecified | 19 | 8.59 [3.36; 20.23] | 96.7 |

before CHIKF was diagnosed as reported in 35/65 studies. Treatment was supportive through management of aches using analgesics, volume resuscitation by intravenous hydration and mechanical ventilation in cases of respiratory distress. Neurological symptoms were managed using anticonvulsant therapy in 12 studies. Abnormal haematological values were corrected by transfusion of either platelets, red blood cells or plasma in 14 studies. Immunoglobulin, antivirals, antimalarial and corticosteroid therapies were also reported (Fig 5). None of these treatments are supported by randomized controlled trials, or by nationally recognized guidelines.

Fifty-five studies did not describe the disease outcome. After pooling all the cases in the included studies (n=3093), 34.5% (n=1068) underwent full recovery within an average of 11 days. 1.2% (n=36) developed a severe form of the disease that necessitated hospitalisation and special care. The severity included long-term neurological and dermatological sequelae, severe cerebral bleeding, neurocognitive delay, cerebral palsy with ataxia and blindness, ocular and behavioural or postural deficiencies (dysconjugate gaze), language delay, axial hypotonia, aphasia, tenosynovitis causing fixed flexion deformity of thumb, hypotonic cerebral palsy with mental retardation, deafness, persistent seizures, visual impairment, tone abnormalities, strabismus, persistent itchy rash, arthralgia, progressive sclerosis of digital skin with limited range of motion and further tapering of distal fingertips, postnatal microcephaly and retinopathy, ptosis and myosis, ataxia, pallesthesia and hyperflexia in lower limbs associated with persistent clonus and urinary incontinence, bulging fontanelle, convulsion and dyspnea. A case fatality rate of 1% (n=33 cases) was observed. The causes of death included respiratory stress, cardiac arrest, intraventricular, gastrointestinal or cerebral haemorrhage, renal failure, cerebral edema, pleural and pericardial effusion, enterocolitis, coma and collapse of the circulatory system.

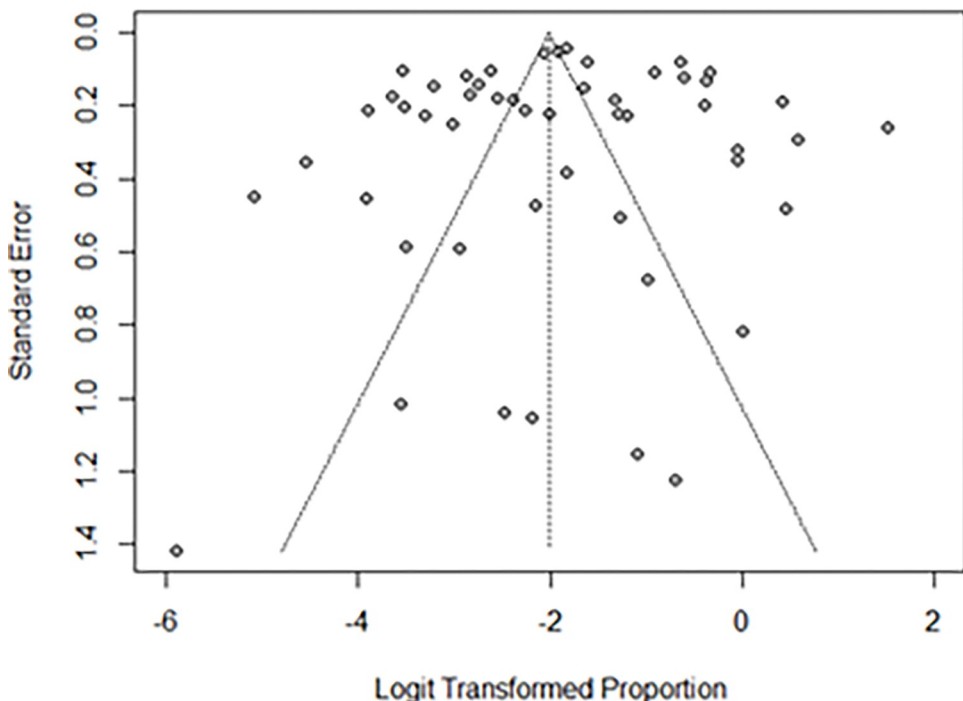

**Fig 3. Funnel plot showing the precision of 53 reviewed studies included in the meta-analysis of CHIKV prevalence among children against their effect estimates.**

## Discussion

In this study, we set out to understand the global epidemiology of CHIKF among children (aged <18 years) by describing its geographical distribution, percentage of positive cases among children presenting with fever, and clinical manifestations. CHIKF is common among febrile children with a wide geographical range of transmission given that it has been reported in five continents. Several studies were from India during the CHIKF epidemic wave in 2005-6 [101–103], and an initial study from the Americas was published after 2013 when CHIKF was first reported in the region [22]. There were relatively few studies published from Africa and Southeast Asia despite a high prevalence of CHIKV in these settings [29].

Spread of CHIKV into naïve populations in the Indian ocean islands, south east Asia and then to the Americas and the Caribbean islands, resulted into rapid establishment of local transmission and high clinical attack rates among children [61, 62, 94]. The intensity of transmission has been associated with disease severity [54]. The CHIKV force of infection and associated clinical manifestations are influenced by the exposure dose [62, 94]. The epidemics resulted into significant absenteeism among school-going children, excessive demand for health care, marked economic losses and significant morbidity and mortality [62]. Neonates were disproportionately affected owing to their developing immune system [62].

The precise prevalence rate of CHIKF among children with acute illness remains undetermined. This is due to scarcity of data from most CHIKF endemic regions. Underreporting of CHIKF cases could be due to exclusion of CHIKF from routine screening of undifferentiated febrile illness in many low- and middle-income countries, absence of affordable diagnostic infrastructure, general lack of awareness among healthcare professionals and absence of an

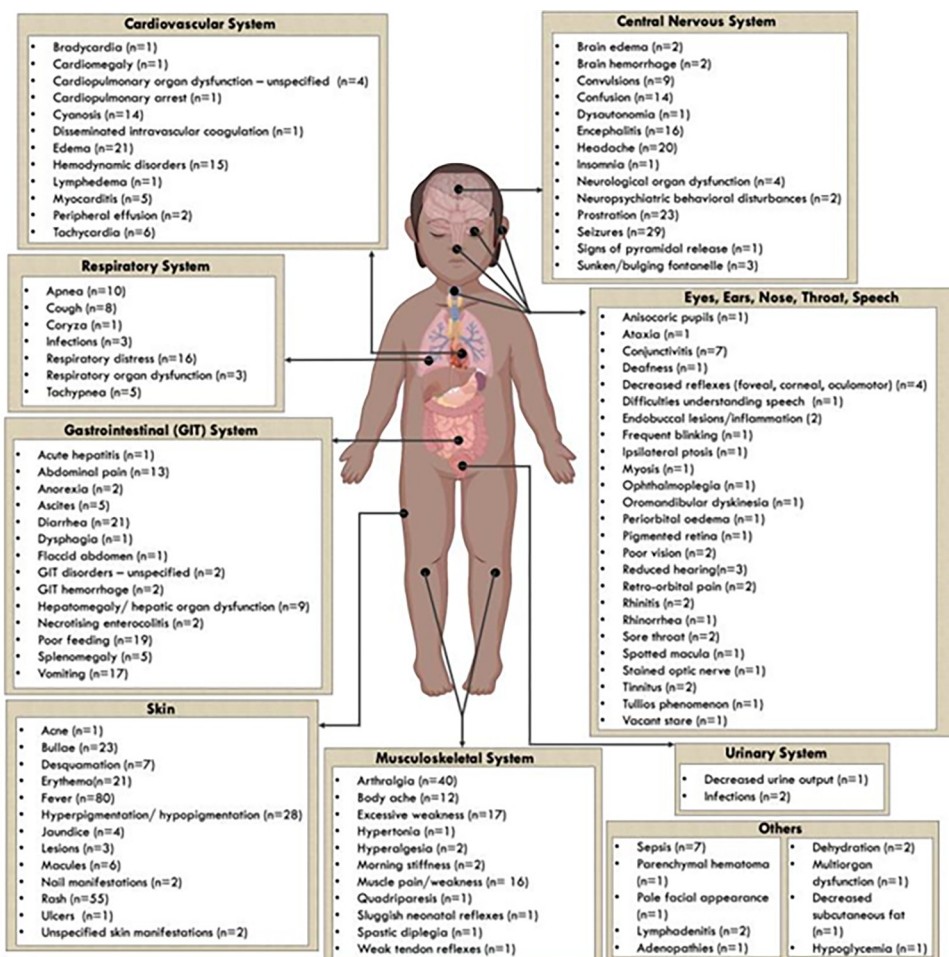

**Fig 4. A summary of symptoms/complications of CHIKV infection in children.** The number in brackets represents the number of studies that reported on this symptom/complication.

efficient surveillance system. Limited epidemiological data on CHIKV among children calls for more efforts towards control, prevention and management within the clinical and public health sector.

There is no specific clinical sign that is discriminatory for CHIKF in children to aid in its diagnosis, and hence CHIKF may be missed in many countries that lack testing capacity. CHIKF manifests with a wide range of clinical symptoms affecting most parts of the body including musculoskeletal, nervous, cardio- respiratory, renal, cutaneous and gastrointestinal systems. Neonates may become infected during the peri-partum period if born to viremic mothers, and in rare cases may develop long-term neuro-cognitive impairment or fatal pathological progression [35]. The rate of vertical transmission among CHIKV-infected pregnant women was estimated at 48.5% during an epidemic [104].

CHIKF clinical manifestations among children vary with geographical location and the infecting CHIKV genotype. CHIKF associated thrombocytopenia in children is dependent on the infecting CHIKV genotype. It was observed more among children infected by the ECSA and IOL genotypes [35, 72, 82,105–]. Clinical investigations of CHIKF among children in the Carribbean island and Central America noted absence of this common hematological parameter reported in most studies from Asia, Indian ocean islands and Africa [34, 62, 68].

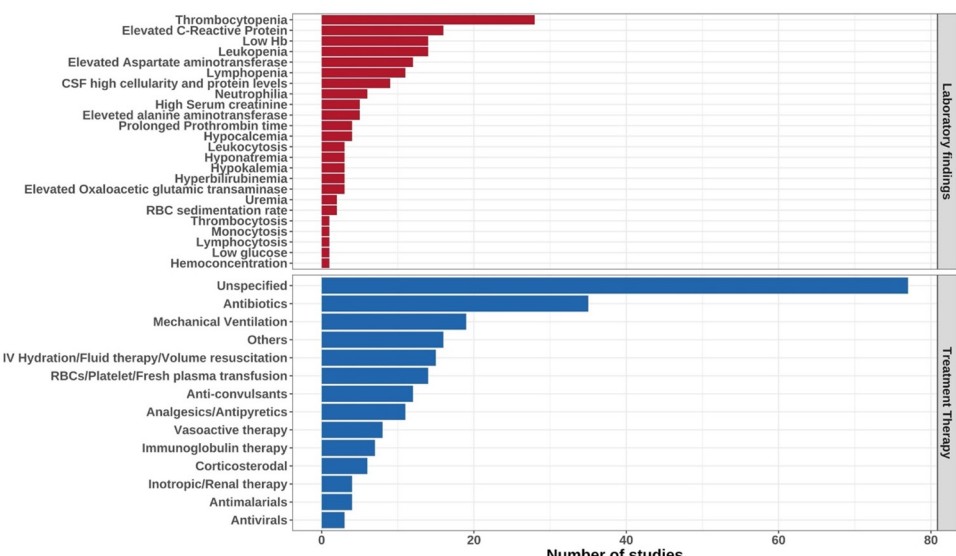

**Fig 5. A summary of the treatments administered to CHIKV infected children and the reported laboratory investigation findings among studies included in this review.**

Surveillance and detection of CHIKV infections among children is rare within community setups as most studies in this review were hospital-based reports. Community surveillance could inform transmission trends and enable timely detection of epidemics. Children travellers from affected areas can contribute in the importation of CHIKV to new niches as indicated by the case reports from Europe [46, 51, 53, 106]. This could be detrimental among CHIKV immunologically naïve populations. An effective surveillance system can detect transmission trends and help in prediction and identification of epidemic seasons.

We recognise some limitations in our review that calls for caution when interpreting our results. Differential representation of various geographical zones in terms of publications could lead to bias. Scarcity of data from endemic regions may lead to an underestimate of the proportion of positive cases in this review, or a bias towards studies during epidemics may lead to an overestimate. We acknowledge that there would be differences/biases based on socio-demographic conditions of different settings which can impact on transmission dynamics and disease severity.

## Conclusions

CHIKF is a significant unrecognised and underreported health problem among children globally and should be included in routine screening of febrile illnesses. Efforts towards treatment therapy and vaccine development should target the vulnerable children <1 year of age who are at an increased risk of developing severe CHIKV-associated neurological complications and require adequate monitoring for potentially fatal outcomes.

## Supporting information

**S1 Checklist. PRISMA criteria for Chikungunya disease among children systematic literature review.**
(DOCX)

**S1 Fig. The distributions of the included studies versus their year of publication.**
(TIF)

**S1 Table. Full details of 142 studies identified by a systematic literature search for Chikungunya infection among children published between 1983 and 2021 inclusive.**
(XLSX)

**S2 Table. Quality appraisal using Joanna Briggs Institute Critical Appraisal tool checklist for 58 studies identified for a systematic literature review of Chikungunya infection among children.**
(XLSX)

**S3 Table. Quality appraisal using National Institute of Health quality assessment tool for 53 longitudinal and cross-sectional studies identified for a systematic literature review of Chikungunya infection among children.**
(XLSX)

**S4 Table. Quality appraisal using National Institute of Health quality assessment tool for 13 case series identified for a systematic litertaure review of Chikungunya infection among children.**
(XLSX)

**S5 Table. Characteristics of the best five models fitted by the multimodel inference in metaregeression.**
(DOCX)

## Acknowledgments

This manuscript was submitted for publication with permission from the Director of the Kenya Medical Research Institute.

## Author Contributions

**Conceptualization:** Doris K. Nyamwaya, Samuel M. Thumbi, George M. Warimwe, Jolynne Mokaya.

**Formal analysis:** Doris K. Nyamwaya, Samuel M. Thumbi, Philip Bejon, George M. Warimwe, Jolynne Mokaya.

**Funding acquisition:** Philip Bejon, George M. Warimwe.

**Investigation:** Doris K. Nyamwaya, Philip Bejon, George M. Warimwe, Jolynne Mokaya.

**Methodology:** Doris K. Nyamwaya, George M. Warimwe, Jolynne Mokaya.

**Project administration:** George M. Warimwe.

**Resources:** George M. Warimwe.

**Supervision:** Samuel M. Thumbi, Philip Bejon, George M. Warimwe, Jolynne Mokaya.

**Validation:** Doris K. Nyamwaya, Samuel M. Thumbi, Philip Bejon, George M. Warimwe, Jolynne Mokaya.

**Writing – original draft:** Doris K. Nyamwaya, Jolynne Mokaya.

**Writing – review & editing:** Doris K. Nyamwaya, Samuel M. Thumbi, Philip Bejon, George M. Warimwe, Jolynne Mokaya.

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
