## [Decision Letter · Decision Letter 0]

15 Jun 2022

PGPH-D-22-00382

The global burden of Chikungunya fever among children: A systematic literature review and meta-analysis

Dear Dr. Nyamwaya

Thank you for submitting your manuscript to PLOS Global Public Health. After careful consideration, we feel that it has merit but does not fully meet PLOS Global Public Health’s publication criteria as it currently stands. Therefore, we invite you to submit a revised version of the manuscript that addresses the points raised during the review process.

Please submit your revised manuscript by . If you will need more time than this to complete your revisions, please reply to this message or contact the journal office at globalpubhealth@plos.org. Please include the following items when submitting your revised manuscript:

We look forward to receiving your revised manuscript.

Kind regards,

Megan Coffee, MD, PhD

Academic Editor

Journal Requirements:

1. Figure 2: please (a) provide a direct link to the base layer of the map used and ensure this is also included in the figure legend; (b) provide a link to the terms of use / license information for the base layer. We cannot publish proprietary or copyrighted maps (e.g. Google Maps, Mapquest) and the terms of use for your map base layer must be compatible with our CC-BY 4.0 license. 

Additional Editor Comments (if provided):

Thank you for addressing Chikungunya. This is a very important topic and one I care very much about.

I would reclassify locations as having a) novel introductions b) epidemics c) endemic d) mix e) none.

The first category has been an important one in ChikV dynamics over the last 15 or so years. As ChikV traveled with its new mosquito friends (Aedes albopictus) to Reunion and Kerala and elsewhere in India, and then on to the Caribbean and Central America and on to Brazil and the rest of South America, we saw dramatic case numbers. Whole communities had ChikV all at once. There are areas where no one living there did not have the "fever". The high viremia with ChikV means it has a very rapid "urban" transmission cycle with a very high rate of symptomatics. After a wave, the "fever" dies out because there are not enough susceptibles given the high attack rate and immunity is long lasting from a single infection (though of course we may see different strains behave differently). If transmission persists, there may be new, but smaller, surges as susceptibles build up and there are enough small children, new migrants, and perhaps some protected persons (away during first surge or in an AC- house).

This dynamic will be very different from an area that has had ChikV for generations where there will be smaller waves as transmission occurs in presumably younger children without exposure to ChikV before, much like measles or chickenpox.

I would not highlight the 11% figure. This will likely be misread and could lead to a misunderstanding about ChikV dynamics. Given all the work that's gone into this paper, I would not want that to happen.

Instead as ChikV crashes as a surging wave in areas without prior immunity, averages, just like we have seen with COVID surges, would not be as meaningful. Instead cumulative prevalence would be interesting, but that's not available. I would instead highlight what endemic rates are, which is important and quite high, and also point to how high these can be in novel outbreaks. I don't think pooling these would help explain this to readers though.

This is much like trying to capture COVID surges and the difficulty of point prevalences. We saw how devastating COVID can be in a population with no immunity and with increase travel and climate/environment change we will presumably see more so this is an important topic being discussed here beyond just ChikV. I would think of this as like when Omicron first caused sudden large surges; it may not be the norm after. Endemic areas the rate would be more meaningful as it has reached more of a steady state, rather than trying to capture a rate of a moving target in a surge.

Moreover age will be a very different factor in an endemic area where children are most likely not to have immunity compared to a novel introduction where all ages have no immunity.

I would also try to bring in some of the information these papers bring about the specifics of the cases in pediatrics. There was a lot of rich information on case presentations in the literature cited here, especially regarding infants and congenital cases. On page 9 and 10 many factors are listed but more discussion would help (especially as some factors are in opposition to some degree like neutrophila and leukopenia). It is helpful that thrombocytopenia is stressed and often is not recognized.

I would also consider reformatting how geographic distributions are discussed. Many of the cases in Europe and North America (outside of the Caribbean and Central American countries) were from travel and I would (though it is mentioned for Europe) carve this would a separate category. Moreover, there are also many in the Caribbean and Central America who would not describe themselves as in North America. Given the epidemiological dynamics of ChikV were so different in Central America and the Caribbean, I might make Central America and Caribbean a separate category (or categories). It would be better understood by people in these areas and also reflect the climate differences that are quite important here.

I might discuss more about the reasons for geographic differences (mosquito change and climate).

Reviewers' comments:

Reviewer's Responses to Questions

**Comments to the Author**

1. Does this manuscript meet PLOS Global Public Health’s publication criteria? Is the manuscript technically sound, and do the data support the conclusions? The manuscript must describe methodologically and ethically rigorous research with conclusions that are appropriately drawn based on the data presented.

Reviewer #1: No

Reviewer #2: Yes

2. Has the statistical analysis been performed appropriately and rigorously?

Reviewer #1: Yes

Reviewer #2: Yes

3. Have the authors made all data underlying the findings in their manuscript fully available (please refer to the Data Availability Statement at the start of the manuscript PDF file)?

Reviewer #1: Yes

Reviewer #2: Yes

4. Is the manuscript presented in an intelligible fashion and written in standard English?

Reviewer #1: Yes

Reviewer #2: Yes

5. Review Comments to the Author

Reviewer #1: The study was carried out on “The global burden of Chikungunya 1 fever among children: A systematic literature review and meta-analysis” and provided a detailed information on Chikungunya and clinical facts in <18 years age group. The study was done systematically and data analyzed properly. The major comments are as follows:

• There is lot of literature is available on Chikungunya fever illness, however authors has not much focused on epidemiology, geographical distribution and pathophysiology of diseases in introduction part.

• Why only <18 age group. What difference was observed from published literature between <18 and >18 groups in Chikungunya clinical conditions.

• Authors should have mentioned a paragraph on differences between <18 and >18 chikungunya clinical as well as treatment conditions.

• Clinical condition between different geographic conditions should be included which might have provide a good knowledge (genetically) further to understand the molecular epidemiology of Chikungunya.

Reviewer #2: This is a well-written review and meta-analysis that underscores the importance of improving surveillance for this relatively neglected viral disease.

My suggestions are minor:

1. Abstract / Findings / line 50-52 -- there are a couple of confusing elements in here and I suggest rewriting to clarify. E.g., the statement "the pooled positivity rate among acutely unwell children was 11%" implies that this was the rate among all unwell children, and the text and tables suggest that the highest rate was from South America (22.8%), not North America (22. 2%), but in any case, these two estimates are so close that perhaps they should be combined to "the Americas".

2. Several minor proofreading suggestions by line number -- please consider whether you wish to accept these:

49: change 'least' to 'fewest'; add 'more' -- "included studies were more commonly conducted"

64: positive-sense

66: change 'could' to 'may'

67: delete colon

74: delete 'within', add 'the' -- "the Pacific Islands"

81: add comma after 'underestimated'

88: subject-verb agreement 'there remains limited descriptions'

91: tested positive 'for CHIKV'

100: capitalize Chikungunya

108: add parentheses (DKN and JM)

109: laboratory-confirmed

110: replace 'those' with 'and'

116: Google Sheets

118-119: what if a study spanned both epidemic and non-epidemic seasons?

131: delete 'and' -- "population that screened participants"

136: superscript the 2 in r2

137: intervals (with an 's')

148: significant

154: delete 'being'

203: leukopenia (to be consistent with spelling elsewhere)

211: please write out numbers at the start of a sentence -- "Sixty-five of one hundred forty-two studies"

231: I believe the English term for 'apalastesis' is pallesthesia

242-244: confusing -- there are 16 included studies from the Americas, so why do the authors refer to one study from the Americas after 2013? I fear I am missing their point.

243: 'relatively' few studies -- because by number, there are quite a few, but it may be that there are few relative to the number of CHIKF cases in these regions

255: replace 'indicates' with 'suggests'?

260: born to viremic mothers

261: CHIKV-infected

263: set-ups, as

269: call

277: delete 'the'

Figure 3: captions refer to "CHIKV cases" which is a bit confusing as I thought the disease was CHIKF and the virus was CHIKV -- please clarify what's meant here by CHIKV cases

6. PLOS authors have the option to publish the peer review history of their article (what does this mean?). If published, this will include your full peer review and any attached files.

**Do you want your identity to be public for this peer review?** For information about this choice, including consent withdrawal, please see our Privacy Policy.

Reviewer #1: No

Reviewer #2: No

---

## [Decision Letter · Decision Letter 1]

12 Sep 2022

PGPH-D-22-00382R1

The global burden of Chikungunya fever among children: A systematic literature review and meta-analysis

Dear Dr. Nyamwaya,

Thank you for submitting your manuscript to PLOS Global Public Health. After careful consideration, we feel that it has merit but does not fully meet PLOS Global Public Health’s publication criteria as it currently stands. Therefore, we invite you to submit a revised version of the manuscript that addresses the points raised during the review process.

We look forward to receiving your revised manuscript.

Kind regards,

Megan Coffee, MD, PhD

Academic Editor

Journal Requirements:

Additional Editor Comments (if provided):

Thank you for this work and all of your edits.

A few points:

1. antibiotics/bacteria

You mention in the abstract "antibiotics for presumed secondary bacterial infection", but there is not much more about antibiotics and secondary bacterial infections in the paper. It is mentioned "234 prescription was predominantly given to children presenting with undifferentiated fever before CHIKF 235 was diagnosed as reported in 35/65 studies" and it is show in the but this is different than treating bacterial superinfections, this is empiric treatment and misdiagnosis. Is there any information on secondary bacterial infections? In my experience, these are rather rare. I might adjust the abstract to match the paper more closely.

2. genotypes and different chikv types

You may want to include a table (or map) showing the differences suspected between genotypes and where they have spread and which mosquitoes involved.

You state in the intro "There are

64 three main CHIKV genotypes, the west African (WA) , the east central and southern African (ECSA)

65 and the Asian genotypes [4]. Mutations in the ECSA genotype resulted in the Indian ocean lineage

66 (IOL) [5] ." I would add more in the intro to flesh this out.

I would avoid saying "It is commonly caused by the ECSA and IOL

294 genotypes" as it makes it seem like ECSA and IOL are different. I might try to explain more if ECSA is thought to be the cause or whether this was seen more in the Indian Ocean but more just try to explain more in the intro about the different genotypes and their associations.

I would add some info on the differences in genotypes outside of pediatrics to intro and leave differences in pediatrics for the end.

I might clarify what milder meant (esp as health surveillance data varies), but could add this sort of info to the intro, except for the thrombocytopenia in children.

Studies in the Carribbean Islands and Central America where outbreaks were caused by the

291 Asian genotype, indicated a milder course of disease [47,48] when compared to those from Reunion

292 islands where the Indian Ocean lineage circulated [85,98] . CHIKF associated thrombocytopenia in

293 children is dependent on the infecting CHIKV genotype.

I would also clarify more about the distribution, as ECSA was introduced into the Americas. https://pubmed.ncbi.nlm.nih.gov/33861753/

The systematic review does in figure 1 explain why studies were included/excluded. You can provide more clarifications as requested by Reviewer 2.

Do try to, as Reviewer 1 states, to "add information about the epidemiology, geographical distribution and pathophysiology of disease"

Reviewers' comments:

Reviewer's Responses to Questions

**Comments to the Author**

1. If the authors have adequately addressed your comments raised in a previous round of review and you feel that this manuscript is now acceptable for publication, you may indicate that here to bypass the “Comments to the Author” section, enter your conflict of interest statement in the “Confidential to Editor” section, and submit your "Accept" recommendation.

Reviewer #1: All comments have been addressed

Reviewer #3: (No Response)

2. Does this manuscript meet PLOS Global Public Health’s publication criteria? Is the manuscript technically sound, and do the data support the conclusions? The manuscript must describe methodologically and ethically rigorous research with conclusions that are appropriately drawn based on the data presented.

Reviewer #1: Yes

Reviewer #3: (No Response)

3. Has the statistical analysis been performed appropriately and rigorously?

Reviewer #1: Yes

Reviewer #3: (No Response)

4. Have the authors made all data underlying the findings in their manuscript fully available (please refer to the Data Availability Statement at the start of the manuscript PDF file)?

Reviewer #1: Yes

Reviewer #3: Yes

5. Is the manuscript presented in an intelligible fashion and written in standard English?

Reviewer #1: Yes

Reviewer #3: Yes

6. Review Comments to the Author

Reviewer #1: 1. Page 12, line 274: Correct the spelling of ‘mortality’.

2. Response to reviewer-1, comment-1, The added information about the epidemiology, geographical distribution and pathophysiology of disease is not sufficient. Must focus on recent literature about global epidemiology and geographical distribution of chikungunya fever.

3. Reviewer-1, comment-3 is not answered sufficiently. Instead the given response to this comment would answer the comment-2 aptly. The authors should include a paragraph highlighting various clinical conditions among different geographic conditions.

Reviewer #3: The manuscript presented may be of interest, however, I have some major issues with how the study was conducted.

1. The statement in the Background section states, "...we conducted a systematic literature review and meta-analysis to determine the epidemiology of CHIKF in children globally...".

I have an issue here since in many parts of the world there are socio-demographic conditions that can heavily impact transmission and even the severity of the disease (access to good health care).

2. In the Findings section, it states that of the 2104 studies only 142 and 53 were included in the systematic literature review and meta-analysis. It is not clear why.

In the conclusions, it says that it is an underreported health problem among children. This is actually true for most (if not all) infectious diseases since young children have a developing immune system.

7. PLOS authors have the option to publish the peer review history of their article (what does this mean?). If published, this will include your full peer review and any attached files.

**Do you want your identity to be public for this peer review?** For information about this choice, including consent withdrawal, please see our Privacy Policy.

Reviewer #1: No

Reviewer #3: No

---

## [Decision Letter · Decision Letter 2]

18 Nov 2022

The global burden of Chikungunya fever among children: A systematic literature review and meta-analysis

PGPH-D-22-00382R2

Dear Miss Nyamwaya,

We are pleased to inform you that your manuscript 'The global burden of Chikungunya fever among children: A systematic literature review and meta-analysis' has been provisionally accepted for publication in PLOS Global Public Health.

Best regards,

Julio Croda, Ph.D, M.D.

Academic Editor

Reviewer Comments (if any, and for reference):

Reviewer's Responses to Questions

**Comments to the Author**

1. If the authors have adequately addressed your comments raised in a previous round of review and you feel that this manuscript is now acceptable for publication, you may indicate that here to bypass the “Comments to the Author” section, enter your conflict of interest statement in the “Confidential to Editor” section, and submit your "Accept" recommendation.

Reviewer #1: All comments have been addressed

Reviewer #3: All comments have been addressed

2. Does this manuscript meet PLOS Global Public Health’s publication criteria? Is the manuscript technically sound, and do the data support the conclusions? The manuscript must describe methodologically and ethically rigorous research with conclusions that are appropriately drawn based on the data presented.

Reviewer #1: Yes

Reviewer #3: Yes

3. Has the statistical analysis been performed appropriately and rigorously?

Reviewer #1: Yes

Reviewer #3: N/A

4. Have the authors made all data underlying the findings in their manuscript fully available (please refer to the Data Availability Statement at the start of the manuscript PDF file)?

Reviewer #1: Yes

Reviewer #3: Yes

5. Is the manuscript presented in an intelligible fashion and written in standard English?

Reviewer #1: Yes

Reviewer #3: Yes

6. Review Comments to the Author

Reviewer #1: (No Response)

Reviewer #3: No further comments.

7. PLOS authors have the option to publish the peer review history of their article (what does this mean?). If published, this will include your full peer review and any attached files.

**Do you want your identity to be public for this peer review?** For information about this choice, including consent withdrawal, please see our Privacy Policy.

Reviewer #1: No

Reviewer #3: No
